# Parental Psychological Flexibility as a Mediating Factor of Post-Traumatic Stress Disorder in Children after Hospitalization or Surgery

**DOI:** 10.3390/ijerph182111699

**Published:** 2021-11-07

**Authors:** Amichai Ben-Ari, Roy Aloni, Shiri Ben-David, Fortu Benarroch, Daniella Margalit

**Affiliations:** 1Department of Behavioral Sciences, Ariel University, Ben-Zakai 36/8, Jerusalem 9318659, Israel; roykoa@gmail.com (R.A.); daniellam@ariel.ac.il (D.M.); 2Herman Dana Division of Child and Adolescent Psychiatry, Hadassah-Hebrew University Medical Center, Jerusalem 91240, Israel; FORTUBEN@hadassah.org.il; 3Department of Psychology, Hebrew University, Jerusalem 9190501, Israel; shiri.bennaim@gmail.com; 4Hadassah Medical Center, Department of Psychiatry, Jerusalem 91120, Israel

**Keywords:** pediatric medical traumatic stress, post traumatic stress disorder, parental psychological flexibility, children after hospitalization

## Abstract

Background: Illness, surgery, and surgical hospitalization are significant stressors for children. Children exposed to such medical events may develop post-traumatic medical syndrome (PMTS, pediatric medical traumatic stress) that could slow their physical and emotional recovery. Objective: This study examined the relationship between the level of parental psychological resilience and the development of PMTS in young children. Method: We surveyed 152 parents of children aged 1–6 who were admitted to the pediatric surgery department. Parents completed questionnaires in two phases. In the first phase, one of the parents completed the Acceptance and Action Questionnaire (AAQ-ll) and the Parental Psychological Flexibility (PPF) Questionnaire. In the second phase, about three months after discharge, the same parent completed the Young Child PTSD (Post Traumatic Stress Disorder) Checklist (YCPC) and the UCLA (Los Angeles, CA, USA) PTSD Reaction Index for DSM-5 Parent/Caregiver Version for Children Age 6 Years and Younger Evaluating Post-traumatic Disorder. In addition, the parent completed a Posttraumatic Stress Diagnostic Scale (PDS) questionnaire to assess the existence of post-traumatic symptoms in the parents. Results: The findings indicate that (1) a parent’s psychological flexibility is significantly associated with the level of personal distress (r = −0.45, *p* < 0.001), (2) a parents’ level of distress is significantly correlated with the child’s level of PTMS, and (3) a parent’s level of psychological flexibility is a significant mediating factor between the level of parental post-traumatic distress and the child’s level of PTMS. Conclusions: A parent’s psychological flexibility may act as a protective factor against the development of the child’s mental distress after hospitalization or surgery.

## 1. Introduction

Although the diagnosis of post-traumatic stress disorder (PTSD), originally formulated in 1980, was not believed to be relevant to children and adolescents, it is now well accepted that children and adolescents can develop PTSD following life-threatening and/or traumatic events [1,2]. Although PTSD in older children and adolescents is quite similar to that of adults, [3] preschool aged children express their distress differently from adults and older children [4]. To reflect that difference, a new PTSD subtype, post-traumatic stress disorder for preschool-aged children (children under six), was included in the DSM-V [5].

Studies have shown that medical interventions can develop into a traumatic experience for children and their families [6,7,8,9], causing PTSD-like symptoms. Illness, injury and the accompanying medical interventions are significant stressors for children who may develop symptoms indicative of post-traumatic stress disorder, such as over-arousal, re-experiencing the event, and developing avoidance patterns. These, in turn, can lead to significant functional impairment [6,8,10,11].

This development is now referred to as Pediatric Medical Trauma Syndrome (PMTS). PMTS has been identified in a variety of pediatric patient populations, including cancer patients [12], surgery patients [6], cardiac surgery patients [13], burn victims [14], and PICU (Pediatric Intensive Care Unit) patients [15].

Studies on the development of PMTS in children after a medical procedure have indicated the following risk factors: parental stress response to the medical event, duration of hospitalization, impairment of memories from hospitalization, severity of hospitalization [11,16], delusional memories or troublesome hallucinations about hospitalization [17], exposure to a high intensity of pain, being alone without a parent or other caregiver, and lack of peer support [18,19]. In addition to these risk factors, the parent’s psychological flexibility has recently been identified as another possible influence on the development of a child’s PMTS [20,21,22].

Psychological flexibility is a broad concept that includes an individual’s ability to identify current situational demands, both external and internal, and adapt accordingly [23]. More specifically, psychological flexibility is the ability of the person to fully experience life while connecting to the present moment without a judgmental position or excessive protection of oneself and those around them [24,25,26,27,28]. Thus, psychological inflexibility is considered a nucleus underlying many psychopathologies and even of general psychological distress [27,28,29,30,31]. Parental resilience is a broad category that includes within it many parameters including intelligence, previous psychopathology, socioeconomic status, and self-worth. In short, flexibility is a component within the spectrum of resilience.

Parental psychological flexibility includes the ability to experience difficult parenting experiences while connecting to the present moment with a non-judgmental attitude toward oneself, the parent, thereby reducing the need to exert control over both their individual experiences and that of the child [32,33,34]. This position produces maladaptive psychological responses that cause the person’s actions to be inconsistent with their values and cause them great harm [28,29]. This is mainly expressed in two configurations: one is the attempt to control the current situation and the other is general avoidance of internal and external experiences [27,29].

A parenting style in which adaptation is lacking [24] can cause problems in a child’s psychological and emotional development and mental well-being. The lack of parental flexibility is associated with practicing avoidance, which may lead to increased parental stress and child psychological problems [30].

Given these findings that the parents’ mental attitude and coping skills in times of stress are a risk factor for the development of PMTS in their children [20,21,22,24], the aim of the present study was to examine the relationship between parental psychological flexibility and PMTS development in children after a medical procedure.

## 2. Materials and Methods

### 2.1. Procedure

The research procedure consisted of two steps. In the first phase, from April to July 2018, 218 parents of children consecutively hospitalized in the pediatric surgery ward during the study period were approached. Parents of children aged 1–6 were included if they were literate in Hebrew and the child had not suffered brain injury or undergone cranial surgery. Of the eligible sample, 184 agreed to participate; 28 parents refused due to lack of time, and six parents refused, stating that they were emotionally overwhelmed. Ethics approval was granted by the sponsoring university hospital (HMO0437-14) and informed consent was given. An explanation of the study was provided to the parents by the study’s investigative team, including the PI and the research assistants.

During the second phase, approximately three months after discharge, the UCLA PTSD and the YCPC questionnaires were administered to measure the development of post-traumatic symptoms in children, and the PDS questionnaire was given to measure the parent’s distress level. In the second phase, 152 parents participated; 32 did not answer the phone, and three refused due to lack of time.

### 2.2. Measures

#### 2.2.1. Demographic Questionnaire

This questionnaire included sociodemographic information, parental religious affiliation, and years of education, a self-report of socioeconomic status, the reason for hospitalization and the type of hospitalization (elective/emergency).

#### 2.2.2. Acceptance and Action Questionnaire (AAQ) 

Self-report questionnaire was used to assess parental psychological flexibility. The questionnaire includes seven items rated on a modified Likert scale of 1–7, one being “never true” and seven being “always true”. Examples of items include: “My painful memories prevent me from living a full life.” or “Worries are an obstacle to my success.” Internal reliability stands at 0.87. After back translation into Hebrew, the internal reliability for the translated questionnaire in the present study was Cronbach α = 0.87. Higher scores indicate a higher level of flexibility [35].

#### 2.2.3. Parental Psychological Flexibility Questionnaire (PPF)

A self-report questionnaire was used for assessing parental psychological resilience. The questionnaire includes 19 items ranked on a modified Likert scale of 1–7, one being “never true” and 7 being “always true”. Examples of items include: “My feelings make it difficult for me to be the ideal parent I would like to be” or “My worries are an obstacle for me to be a successful parent”. The internal reliability of the original study was Cronbach α = 0.90. After back translation into Hebrew, the internal reliability for the translated questionnaire in the present study was Cronbach α = 0.87. Higher scores indicate a higher level of flexibility [36].

#### 2.2.4. UCLA PTSD Reaction Index for DSM-5 Parent/Caregiver Version for Children Aged 6 Years and Younger 

This instrument identifies the presentation of PTSD in young children and includes 16 items, each rated as no = 0, yes = 1. After back translation into Hebrew, the internal reliability for the translated questionnaire in the present study was Cronbach α = 0.95. Higher scores indicate a higher level of distress [37,38].

#### 2.2.5. Young Child PTSD Checklist (YCPC) 

A 42-item parent-report questionnaire examining PTSD among preschool-aged children was used. The first 13 items examine the type of traumatic event the child has undergone. The next 23 items address symptoms and the last section includes six items on functional impairment. After back translation into Hebrew, good internal reliability scores for translated symptom scales were identified: arousal: Cronbach’s α = 0.92; avoidance: α = 0.93; reliving: α = 0.88; total score: α = 0.97. Good internal traceability for the functioning scale was also found: α = 0.96. Higher scores indicate a higher level of distress [39].

#### 2.2.6. Post-Traumatic Stress Diagnostic Scale (PDS)

The PDS49 is a self-report questionnaire used to assess parental post-traumatic distress in parents. The tool includes 48 items rated on a Likert scale of 0–3. The PDS is widely used in clinical and research settings and has a reported internal consistency of a = 0.92. We used the Hebrew translated and validated version [42]. In the current study, internal reliability yielded α = 0.97 [40,41].

### 2.3. Data Analysis

Data were analyzed using SPSS (v. 27, IBM, Chicago, IL, USA). Descriptive statistics were produced using frequencies for categorical variables and means with standard deviations for continuous variables. We assessed associations between variables by using Pearson correlation and then performed hierarchical multivariate regressions to predict trauma severity, medical phobia, and adherence to medical treatment. Finally, to assess the mediation model, we used structural equation modeling (SEM). The following indices were used to evaluate the model: chi-squared, which is acceptable when the value is not significant; the goodness of fit index (GFI), the comparative fit index (CFI), and the non-normed fit index (NNFI), (adequate values—above 0.90, excellent fit—above 0.95); and the root mean square error of approximation (RMSEA) (adequate values ≤0.08; excellent fit ≤0.06) [43]. The level of significance (*p*-value) was ≥0.05.

Two separate instruments were used to measure PMTS in children (YCPC and UCLA) and parental flexibility (AAQ and PPF), to enable comparison and correlation between different measures, and to validate two different ways of measuring both PMTS and parental flexibility to create similar results.

## 3. Results

One hundred fifty-two children and their parents were included in the study, 94 boys (61.8%), 58 girls (38.2%), 61 fathers (41%) and 91 mothers (59%). Parents’ ages ranged from 24 to 51 (M = 34.29, SD = 7.2) and children’s ages ranged from one to six years (M = 2.89, SD = 1.51). Of the children, 103 (68.2%) had been hospitalized for elective surgery (Table 1), and most surgeries were considered to be of low complexity. Hospitalization duration ranged from 1 to 37 days (M = 4.71, SD = 5.70). Demographic and hospitalization data (gender, reason for admission, length of hospitalization) of the sample group were similar to those of the overall population of this hospitalized age group (*N* = 6231) during 2018 in the study medical center. Consequently, the research sample was assumed to be representative of the general pediatric ward population.

The Pearson correlations demonstrate a significant correlation between the presence of child PMTS (using the YCPC score) with parental psychological flexibility (AAQ) (−0.45, *p* < 0.001) and parental post-traumatic distress (0.72, *p* < 0.001). Similar and significant results were found when using the UCLA score and the PFF measurement for parental flexibility (Table 2) The mediation model presented in Figure 1 was tested using bootstrapping analysis. The unstandardized path coefficients indicating the effects of parental psychological flexibility (PPF) on parental post-traumatic distress (path a), the effects of parental post-traumatic distress on child PMTS (path b), and the direct effect of parental psychological flexibility on child PMTS (path c) are presented in Table 3. Parental PDS also significantly mediated the relationship between parental psychological flexibility and child PMTS. In addition, a high correlation was found between the two questionnaires that examined the child’s post-traumatic symptoms (UCLA and YCPC) and the two questionnaires that examined the level of parental psychological flexibility (AAQ and PPF).

## 4. Discussion

We identified that parental psychological flexibility contributed to the mediation of both parental post-traumatic distress and trauma symptoms in children. These findings reinforce the current understanding that a parent’s mental state has a significant impact on the child’s ability to cope with stress and the level of post-traumatic distress [21,24,44]. The main significance of these findings is that parental psychological flexibility can be a resilience factor in preventing or modulating trauma symptoms among young children after a traumatic medical event.

The fact that the level of parental distress at the time of hospitalization was a mediating factor in the relationship between the level of parental psychological flexibility and the level of pediatric PMTS is significant. Many studies point to the relationship between psychological flexibility and mental distress [29,31,45], as well as the causal relationship between parental mental distress and post-symptom development. This relationship is especially relevant to traumatic experiences in children after hospitalization or surgery [6].

When parents become overly anxious, potential risks can be exaggerated and transmitted to the child [22]. Parental distress has long been known to affect children’s distress levels [46]. Children whose parents display anxiety during induction, separation, and the overall hospital stay experience greater anxiety compared to those with “calm” parents [47,48]. In addition, a recent study on slightly older children (8–10 years old) found a direct relation between the mother’s anxiety and children’s traumatic stress levels [49].

Family studies have examined various parenting practices, in a routine home environment, and the resulting impact (protective or harmful) on child and adolescent adjustment [50]. Compared with normative situations of families [51], when parents cope with non-normative devastating situations (e.g., illness, surgery, and hospitalization), the parents’ mental health can be affected. Thus, the focus on parents in family studies should include coping strategies (e.g., parental flexibility) and their impact (positive or negative) on parental competence and adjustment. The current study aims to expand the knowledge base regarding this impact.

The effect of parent psychological flexibility on the levels of parental distress indicates the necessity for parental psychological flexibility during periods of stress. This imperative is supported when looking at the results of studies that have examined the effectiveness of psychological treatment using what is known as an acceptance and commitment therapy, ACT, approach, an orientation to psychotherapy that is based on functional contextualism as a philosophy and RFT as a theory. As such, it is not a specific set of techniques, but rather ACT protocols target the processes of language that are hypothesized to be involved in psychopathology and its amelioration. This can, in turn, reduce distress levels and psychopathological symptoms [26,28,52,53]. Studies examining the effectiveness of ACT treatment in a variety of disorders have found that treatment significantly increased psychological flexibility in patients, leading to symptomatic reduction, which, in turn, reduced distress and increased parental mental well-being [20,28,52,53,54]. In these studies, the effect of treatment translated into an increase in the levels of psychological flexibility among the subjects, adding a significant benefit at the symptomatic level. As many children are hospitalized during early childhood, and because there is difficulty in finding effective interventions for working with children at this age [8,9,10], it is possible that a therapeutic intervention in this approach, focusing on working with parents to improve their psychological resilience, can improve children’s distress. As such, ACT treatment can function as a preventative intervention that can reduce the risk of developing PTMS in children.

The present study has several limitations. The data were from one medical center and included only a population of Hebrew speakers only. Additionally, because the parent’s level of flexibility was measured at the time of the child’s hospitalization, the measurement may have been affected by the stressful situation and, therefore, may not be a reliable indication of the parent’s normal level of flexibility. It is possible that the parent’s flexibility measurement during the child’s hospitalization was affected by the parent’s stress level, yet there is still evidence that a parent’s flexibility is very important in reducing children’s distress after hospitalization or surgery, thus making it important to measure, even during the hospitalization itself.

A significant strength of the study is that measurements were made at two time points, enabling us to measure the development of the processes and produce a causal model that relies not only on the statistics of a mediation model but on the measurement of flexibility before distress and post trauma. Future focus on multisite and multilingual populations would expand the external validity of our findings.

The study findings show that parental psychological flexibility reduces both parental post-traumatic distress and the child’s PTMS. It also shows that parental post-traumatic distress affects the degree of childhood PTMS, and the degree of parental traumatic distress mediates the relationship between parental flexibility and PTMS. Fostering psychological flexibility among the parents of hospitalized children may help develop a resilience factor that could prevent or mitigate PMTS. Thus, parents may benefit from ACT-based preventative interventions that foster present moment consciousness and value-driven behavior, allowing them to better maintain a nonjudgmental awareness of their own experiences so that they are better able to act in accordance with their parenting values. By utilizing such techniques, parents can more effectively engage in adaptive and non-reactive parenting practices (i.e., empathetic listening) that could help the child better cope with the possible trauma of a significant medical event.

## Figures and Tables

**Figure 1 ijerph-18-11699-f001:**
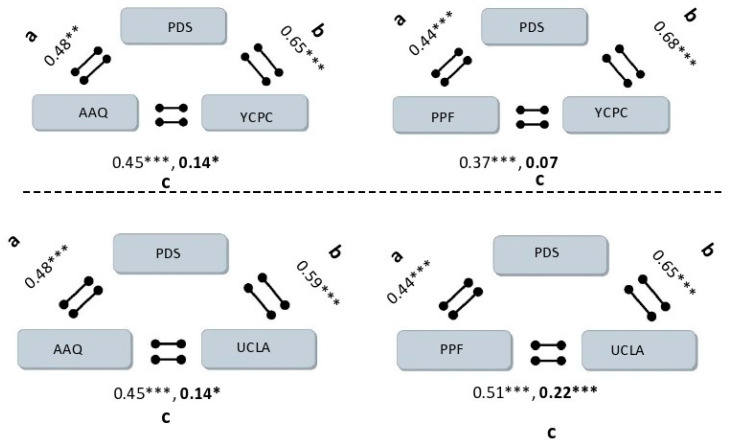
Mediation models with paths a, b, and c; * *p* < 0.05, ** *p* < 0.01, *** *p* < 0.001. The bold values represent the low correlations in the model.

**Table 1 ijerph-18-11699-t001:** Demographic and clinical background characteristics of the sample.

Variable	Mean	* SD	*N*	%
Gender (Males)			94	61.8
Age	2.89	1.52		
Surgery complexity				
Low			91	64.1
Moderate			46	32.4
High			5	3.5
Hospitalization duration	2.60	2.00		
Socio-Economic status				
Low			34	22.2
Moderate			93	61.5
High			25	16.3
Child’s mental difficulties	1.21	0.25		
Parent’s mental difficulties	0.47	0.58		

* Standard deviation.

**Table 2 ijerph-18-11699-t002:** Descriptive statistics and Pearson correlations.

	M	SD	YCPC	UCLA	AAQ	PPF
YCPC (child PTSD measure)	0.34	0.42	-			
UCLA (child PTSD measure)	3.31	3.76	0.85 *	-		
AAQ (parental flexibility measure)	36.75	12.25	−0.45 *	−0.61 *	-	
PPF (parental flexibility measure)	105.60	30.78	−0.37 *	−0.51 *	0.75 *	-
PDS (parental PTSD measure)	2.51	2.57	0.72 *	0.75 *	−0.48 *	−0.44 *

* *p* < 0.001. M- Median, SD- Standard deviation, YCPC- Young Child PTSD Checklist, UCLA-. UCLA PTSD Reaction Index for DSM-5 Parent/Caregiver Version for Children Aged 6 Years and Younger, AAQ- Acceptance and Action Questionnaire, PPF- Parental Psychological Flexibility Questionnaire.

**Table 3 ijerph-18-11699-t003:** Parental distress mediates the relationship between parental psychological flexibility and children’s PTSD—indirect effects and bootstrapping CIs.

Parent Flexibility Measure	Child PTSD Measure	B (SE)	95% CI
AAQ	YCPC	0.01 (0.00)	0.01; 0.01
AAQ	UCLA	0.09 (0.02)	0.05; 0.12
PPF	YCPC	0.00 (0.00)	0.00; 0.01
PPF	UCLA	0.03 (0.01)	0.02; 0.05

PTSD-Post-Traumatic Stress Disorder, B (SE)- Standard error of unpaired estimate, CI- Confidence interval, YCPC- Young Child PTSD Checklist, UCLA-. UCLA PTSD Reaction Index for DSM-5 Parent/Caregiver Version for Children Aged 6 Years and Younger, AAQ- Acceptance and Action Questionnaire, PPF- Parental Psy-chological Flexibility Questionnaire.

## Data Availability

The data that support the findings of this study are available on request from the corresponding author. The data are not publicly available due to their containing information that could compromise the privacy of the research participants.

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
