# Peer review of "Parental Psychological Flexibility as a Mediating Factor of Post-Traumatic Stress Disorder in Children after Hospitalization or Surgery"

_ijerph, 2021, doi:10.3390/ijerph182111699_

Round 1

Reviewer 1 Report

I included comments below:

I found the article interesting, important, and generally well-written. My biggest point is grammatical errors. I am not native language speaker, but I see a number of errors.

My suggestions:

Title:   PTSD ?  I would not recommend using abbreviations in the title of the article.

Abstract:

Line 16-17:  “We 152 parents of children ages of 1-6 who were admitted to our pediatric 16 surgery department at.“ This sentence is not clear and grammatically not correct.

Line 23-27: Missing articles: “the level of personal distress”, “the parent's level of distress”, the level of parental post-traumatic distress”

Introduction

Line 33: PTSD: I have not found meaning of this abbreviation in the text before.

Line 44: may be “a significant functional impairment”

Line 48: may be “in a variety of”

Line 53: “the severity of hospitalization”

Material and methods

Line 97: “questionnaires were administered to measure”

Line 104: “the reason for”

Line 118: ” the internal reliability”

Line 123: “the presentation”

Line 135: “that we used”

Line 146 “structural equation modeling” should without “the “

Line 151: “the level of significance”

Line 152-153: “Two separate instruments were used to measure PMTS in children (YCPC and UCLA) and parental flexibility to compare”

Results

Line 167: “Pearson correlations demonstrate a significant correlation between the presence of child...“

Discussion

Line 189: „ These findings reinforce the current understanding that...“

Line 196: „the level of pediatric PMTS is significant“

Line 205: „In addition, a recent study on...“

Line 208: „The effect of parent psychological flexibility on the level of parental distress...“

Line 227: „...working with parents to improve their psychological resilience can improve...“

Line 230: „The present study has several limitations“

Line 236:  “…yet there is still evidence that parents are very important...“

Line 252: „...allowing them to better maintain nonjudgmental awareness of their experiences so that they are better able to act following their parenting values...“

Reviewer 2 Report

Thank you for an important study, which demonstrates a significance of parent's psychological flexibility, which could be recognized as a potential protective factor against the development of post hospitalization and post-surgery children's mental distress.

However, wider studies including a higher number of patients are suggested in the future.   Also, more attention to a patient’s specific parameters of the autonomic nervous system during PTSD (such as changes of the skin conductance, heart rate, blood pressure, etc.), and its correlation with parental flexibility influence, could be great to consider as well.   It is well known that the neurobiological systems that regulate stress responses are regulated via neurotransmitter pathways in the regions of the brain which are involved in a regulation of conscious and unconscious fear behavior.

Future MRI brain studies in patients with pre- and post PTSD conditions, as well as a measurement received after the parents influence on children post-hospital and post-surgery trauma, could be a valuable addition for the future developments of this research project.

Author Response

Thank you for bringing these important ideas to our attention, including information about patient’s specific parameters of the autonomic nervous system during PTSD as well as MRI brain studies as important elements to include in our future studies.

Reviewer 3 Report

Present manuscript seems to examine the relationship between parental psychological flexibility, post-traumatic stress disorder in children, and posttraumatic stress disorder in parents. Participants were 152 parents (range of age ??, M = ??, SD = ??) and their children (1-6 years, M = 2.89, SD = 1.51). Parental psychological flexibility seems to have two measures: (i) Acceptance and Action Questionnaire (Bond et al., 2011) and (ii) Parental Psychological Flexibility Questionnaire (Burke & Moore, 2015). Post-traumatic stress disorder in children has two measures: (i) UCLA PTSD Reaction Index for DSM-5 Parent (Steinberg et al., 2013; Kaplow et al., 2020), and (ii) Young Child PTSD Checklist (Scheeringa et al., 2013). Post-traumatic stress disorder in parents has been measured with the Posttraumatic Stress Diagnostic Scale (PDS) (Foa et al., 1995, Foa et al., 2016). Overall, as expected, the two measures of parental psychological flexibility have a greater significant positive correlation (.85, p < .001). The same is true for the two measures of post-traumatic stress disorder in children (.75, p < .001). Additionally, a negative relationship was found between parental psychological flexibility and post-traumatic stress disorder (in children and parents).

  • Introduction

The concept parental resilience and its difference with parental flexibility should be explained in the introduction. Additionally, its measure seems to be for parental flexibility.

Authors ought to rationalize a little bit more the role of parents during parental socialization time as well as the consequences of adverse events. Parents have a main responsibility: Raising their children, care them and provide protection against any threat than can affect their healthy development (Hernández-Serrano et al., 2021; Sacca et al., 2021). Parents are always concerned about their children, but especially in the years of parental socialization, during their children's childhood and adolescence (Garcia, Fuentes, Gracia, Serra, & Garcia, 2020). Family studies usually examined the different parenting practices used by parents and its impact (protective or harmful) on child and adolescent adjustment (Perez-Gramaje, Garcia, Reyes, Serra, & Garcia, 2020). Nevertheless, compared to normative situations of families (Fuentes, Alarcon, Garcia, & Gracia, 2015) when parents cope non-normative devastating situations (e.g., illness, surgery and surgical hospitalization), their mental health can be affected, so the focus on parents in family studies should include the coping strategies (e.g. parental flexibility) and its impact (positive or negative) on parental competence and adjustment.

  • Empirical part

In the participants section it is stated that 152 parents were part of the study, but no demographic data such as age are mentioned for parents. There is also no information on statistics such as mean and standard deviation. Besides, in the sample, the country of participants should be included.

Other points:

* It is practically impossible for potential readers follow the reading, in part due to the use of acronyms (e.g., YCPC, UCLA, AAQ, PPF, PDS, etc.). Authors should use the name of the variables

* Authors should use impersonal statements, rather than others as *our pediatric surgery department*

References

Fuentes, M. C., Alarcon, A., Garcia, F., & Gracia, E. (2015). Use of alcohol, tobacco, cannabis and other drugs in adolescence: Effects of family and neighborhood. Anales de Psicología, 31, 1000-1007. doi:10.6018/analesps.31.3.183491

Garcia, O. F., Fuentes, M. C., Gracia, E., Serra, E., & Garcia, F. (2020). Parenting warmth and strictness across three generations: Parenting styles and psychosocial adjustment. International Journal of Environmental Research and Public Health, 17(7487), 1-18. doi:10.3390/ijerph17207487

Hernández-Serrano, O., Gras, M. E., Gacto, M., Brugarola, A., & Font-Mayolas, S. (2021). Family climate and intention to use cannabis as predictors of cannabis use and cannabis-related problems among young university students. International Journal of Environmental Research and Public Health, 18(9308), 1-15. doi:10.3390/ijerph18179308

Perez-Gramaje, A. F., Garcia, O. F., Reyes, M., Serra, E., & Garcia, F. (2020). Parenting styles and aggressive adolescents: Relationships with self-esteem and personal maladjustment. European Journal of Psychology Applied to Legal Context, 12, 1-10. doi:10.5093/ejpalc2020a1

Sacca, L., Rushing, S. C., Markham, C., Shegog, R., Peskin, M., Hernandez, B., Gaston, A., Singer, M., Trevino, N., Correa, C. C., Jessen, C., Williamson, J., & Thomas, J. (2021). Assessment of the reach, usability, and perceived impact of "talking is power": A parental sexual health text-messaging service and web-based resource to empower sensitive conversations with American Indian and Alaska native teens. International Journal of Environmental Research and Public Health, 18(9126), 1-15. doi:10.3390/ijerph18179126

Author Response

Please see attachment. Thank you for your important comments and additional sources that truly clarified points in our manuscript.

Reviewer 4 Report

I would like in future adolescent mcmi scales

of dr millon and adults have millon scales 

/ mbti flexibility must correlate with compulsive  personality 

and j on mbti ! also possibly narcissistic parents can't adapt 

I'm not sure these scales are sufficient to guide 

parents but the academy of pediatrics is without children a scales 

Author Response

Dear Reviewer,

I apologize but I do not understand your comments. I would be happy to make changes if I could understand better your suggestions.

Thank you in advance

Round 2

Reviewer 3 Report

Despite the modifications, some important points should be addressed still to improve the poor quality of the manuscript. All these points should be adequately included to connect previously literature to a correct rationalization of the present study considering classical and current research within the text.

* It is quite confused too the difference between parental flexibility and parental resilience. It should be explained more detailed in the introduction.

*More details should be added in the introduction to rationalize the focus of the article based on the literature on parental socialization:

* Parents have as their primary responsibility the care and protection of their children (Fuentes et al., 2020). For this reason, if their children are faced with an experience in which their health is seriously endangered (e.g., surgery), parents may experience great discomfort and anxiety because it is due to causes external to them (Garcia et al., 2020). In addition, these parental stress may have repercussions on the children's adjustment, but not always the same depending on coping strategies of parents.

* For potential readers it is quite confused due to the writing is focused in the measures rather than in the variables. Therefore, authors should refer along the text to the labels of the variables (parental flexibility, parental resilience, and so), but not to the measures.

* Additionally, it is quite strange why the variable parental resilience is captured with a measure for parental flexibility. Specifically, the measure for the variable parental resilience is captured though the measure Parental Psychological Flexibility (PPF) Questionnaire (Burke & Moore, 2015), “a parent-report measure designed to assess psychological flexibility among parents of pre-adolescents and adolescents (aged 10–18 years). Psychological flexibility within parenting refers to parents’ accepting negative thoughts, emotions and urges about one’s child and still acting in ways that are consistent with effective parenting” (p. 548). Additionally, the questionaire seems usually used as a measure for parental flexibility, but not for parental resilience (e.g., Coyne et al., 2020).

* It should be reviewed adequately the writing of the statistics along the text. For example, were said M 34.29, SD = 7.2 should said M = 34.29, SD = 7.2, with statics (M and SD) in italics.

References

Burke, K., & Moore, S. (2015). Development of the parental psychological flexibility questionnaire. Child Psychiatry & Human Development, 46(4), 548-557. https://doi.org/10.1007/s10578-014-0495-x

Coyne, L.W., Gould, E.R., Grimaldi, M. et al. First Things First: Parent Psychological Flexibility and Self-Compassion During COVID-19. Behav Analysis Practice (2020). https://doi.org/10.1007/s40617-020-00435-w

Fuentes, M. C., Garcia, O. F., & García, F. (2020). Protective and risk factors for adolescent substance use in Spain: Self-esteem and other indicators of personal well-being and ill-being. Sustainability, 12(15), 1–17. https://doi.org/fhd3

Garcia, O. F., Fuentes, M. C., Gracia, E., Serra, E., & Garcia, F. (2020). Parenting warmth and strictness across three generations: Parenting styles and psychosocial adjustment. International Journal of Environmental Research and Public Health, 17(7487), 1-18. https://doi.org/10.3390/ijerph17207487
